# RETRIEVAL OR GLOBAL CONTEXT UNDERSTANDING? ON MANY-SHOT IN-CONTEXT LEARNING FOR LONG-CONTEXT EVALUATION

## ABSTRACT

Language models (LMs) have demonstrated an improved capacity to handle long-context information, yet existing long-context benchmarks primarily measure LMs' retrieval abilities with extended inputs, e.g., pinpointing a short phrase from long-form text. Therefore, they may fall short when evaluating models' global context understanding capacity, such as synthesizing and reasoning over content across input to generate the response. In this paper, we study *long-context language model (LCLM) evaluation* through *many-shot in-context learning (ICL)*. Concretely, we identify the skills each ICL task requires, and examine models' long-context capabilities on them. We ask the first question: *What types of ICL tasks benefit from additional demonstrations, and are these tasks effective at evaluating LCLMs?* We find that classification and summarization tasks show notable performance improvements with additional demonstrations, while translation and reasoning tasks do not exhibit clear trends. This suggests the classification tasks predominantly test models' retrieval skills. Next, we ask: *To what extent does each task require retrieval skills versus global context understanding from LCLMs?* We develop metrics to categorize ICL tasks into two groups: (i) **retrieval** tasks that require strong retrieval ability to pinpoint relevant examples, and (ii) **global context understanding** tasks that necessitate a deeper comprehension of the full input. We find that not all datasets can effectively evaluate these long-context capabilities. To address this gap, we introduce a new many-shot ICL benchmark, **MANY-ICLBENCH**, designed to characterize LCLMs' retrieval and global context understanding capabilities separately. We benchmark 11 open-weight LCLMs using MANYICLBENCH. We find that while state-of-the-art models demonstrate satisfactory performance up to 64k tokens in retrieval tasks, many models experience significant performance drops at only 16k tokens in global context understanding tasks.[1]

## 1 INTRODUCTION

Long-context language models (LCLMs) have revolutionized the way users interact with language models by extending the context size from 2K to 128K or even 1M tokens (Team et al., 2024a; GLM et al., 2024; Dubey et al., 2024), which unlock challenging applications, such as long- and multi-document summarization, multi-turn dialogue, and code repository comprehension. Despite the recent progress in building LCLMs, existing benchmarks primarily evaluate these models' retrieval capabilities (Liu et al., 2023; Hsieh et al., 2024). From synthetic tasks such as Needle-in-A-Haystack (Kamradt, 2023) and RULER benchmark (Hsieh et al., 2024) to real-world challenges like long-novel QA (Karpinska et al., 2024), the majority of benchmarks assess how well LCLMs retrieve specific pieces of information from extensive contexts. As a result, **evaluating models' global understanding of the full context remains lacking**.

To fill the gap, Li et al. (2024) introduce LongICLBench, which uses many-shot ICL classification tasks to evaluate models' long-context performance, arguing that these tasks require the comprehension of the entire input. A few other works have also explored many-shot ICL for long-context

---

[1]Data and code are available at `https://github.com/launchnlp/ManyICLBench`

models (Agarwal et al., 2024; Bertsch et al., 2024). Yet, they have mainly relied on classification tasks (Li et al., 2024; Bertsch et al., 2024), which are insufficient to distinguish which skills LCLMs require to perform well on many-shot ICL classification tasks. Recently, Agarwal et al. (2024) study non-classification ICL tasks but only on Gemini 1.5 Pro. In this work, we want to conduct a comprehensive study on many-shot ICL across a wide range of models, with a goal of identifying tasks that **benefit from additional demonstrations** and explore their utility in evaluating long-context models. Moreover, we seek to determine the extent to which these tasks rely on **retrieval versus global context understanding**.

**RQ1: Which tasks benefit from many-shot ICL?** First, we investigate ICL tasks that are used in prior work, including classification, summarization, and reasoning, under many-shot settings with context lengths from 1k to 128k (Agarwal et al., 2024). We find that classification and summarization tasks show *strong positive correlation between context lengths and model performance*. Our findings indicate that translation and reasoning tasks such as ARC (Clark et al., 2018) and FLORES-200 (NLLB Team, 2022) do not gain much performance with an increasing number of demonstrations. Science and symbolic reasoning tasks exhibit inconsistent trends between context lengths and model performance. This variance in performance is mainly attributed to the specific nature of tasks, where more demonstrations do not boost the models' task understanding. Interestingly, math tasks benefit from additional demonstrations only when step-by-step solutions are derived and using strong LCLMs.

**RQ2: What skill does each task primarily measure?** We then analyze the retrieval and global context understanding skills necessary for each ICL task. We use the ratio between the performance change of removing dissimilar examples and the change of removing similar examples. A high ratio means a more pronounced drop in performance upon removing similar examples, which indicates the task's heavy reliance on retrieval capabilities. Our analysis indicates that existing many-shot ICL *classification* tasks (Li et al., 2024) *predominantly assess retrieval abilities* rather than global context understanding. This leads us to categorize tasks into retrieval and non-retrieval groups.

Subsequently, we explore whether non-retrieval tasks genuinely benefit from additional demonstrations and assess models' global context understanding skills. By comparing the performance of models with unique demonstrations versus duplicated examples on non-retrieval tasks, we aim to determine if duplicating examples adversely affects performance compared to adding new examples. If this is the case, it signifies that unique demonstrations provide additional beneficial information, reinforcing the notion that these tasks require global context understanding. Using this method, we identify a subset of non-retrieval tasks that evaluate models' comprehension of global content.

Following the categorization, we propose a new many-shot ICL benchmark, **MANYICLBENCH**, designed for evaluating long-context models and advocate for the inclusion of many-shot ICL tasks as effective evaluation candidates. Importantly, on MANYICLBENCH, models are tested to either retrieve the most similar demonstrations or assimilate all demonstrations to enhance their understanding of the task (Lin & Lee, 2024; Bertsch et al., 2024). Therefore, MANYICLBENCH *evaluates both retrieval skills and global context understanding*, thus providing a holistic assessment of long-context models' capabilities.

In summary, we make the following contributions in this paper:

- Investigate whether ICL tasks benefit from additional demonstrations and assess their suitability for evaluating LCLMs with a context length up to 128k tokens.

- Develop methods to characterize the primary skills evaluated by ICL tasks, where we focus on distinguishing between retrieval capabilities and global context understanding.

- Construct a many-shot ICL benchmark, named MANYICLBENCH, designed for evaluating LCLMs on both retrieval and global context understanding, while excluding irrelevant datasets previously used in LCLM evaluation.

- Benchmark 11 widely-used state-of-the-art LCLMs on MANYICLBENCH to assess their performance comprehensively.

## 2 RELATED WORK

### 2.1 LONG-CONTEXT LANGUAGE MODELS AND EVALUATION

As large language models grow in scale, there is an increasing demand for handling tasks that require extended contexts. Tasks such as long document summarization (Kryściński et al., 2022), conversations with long-context memory (Xu et al., 2021), and repository-level code completion (Zhang et al., 2023) have garnered significant interest. Advances in efficient attention mechanisms, such as flash attention (Dao et al., 2022) and grouped query attention (Ainslie et al., 2023), alongside the development of GPUs with larger memory capacities, have enabled LLMs to be trained on extended contexts. Techniques like position interpolation (Chen et al., 2023; Peng et al., 2023) and context compression (Chevalier et al., 2023; Mohtashami & Jaggi, 2023; Jiang et al., 2024) have further extended the context window size to up to 1 million tokens.

Despite these advancements, the NLP community still seeks a universal and effective method for evaluating long-context models. One prominent task is Needle-in-a-Haystack (Kamradt, 2023), which requires models to retrieve the most relevant document from a large set of documents. Currently, most evaluation benchmarks focus on synthetic tasks that primarily assess the retrieval capabilities of long-context models (Hsieh et al., 2024; Kamradt, 2023; Lee et al., 2024; Lei et al., 2024). Only a few benchmarks, such as Karpinska et al. (2024) and Zhang et al. (2024), emphasize the model's ability to comprehend the global context. For example, Karpinska et al. (2024) manually curated a set of challenging questions based on various novels to evaluate global context understanding. It is the first work to create a realistic long-context benchmark emphasizing retrieval and global context understanding skills.

### 2.2 MANY-SHOT ICL WITH LCLMS

Because the context length of large language models expands, the number of demonstrations that can be utilized in ICL has also increased. Studies by Li et al. (2024), Bertsch et al. (2024), and Agarwal et al. (2024) have examined various properties of ICL under the many-shot setting. Bertsch et al. (2024) explore whether models are merely performing retrieval tasks or genuinely understanding the tasks during many-shot ICL classification. Similarly, Agarwal et al. (2024) analyzes the performance of tasks beyond classification in the many-shot context, using Gemini-Pro, and finds that additional demonstrations generally enhance task performance. Furthermore, Li et al. (2024) propose a long-context evaluation benchmark LongICLBench comprising many-shot ICL classification tasks, noting that current long-context models still face challenges in this area. None of the prior works has studied what skill each ICL task measures LCLMs for. LongICLBench mostly focuses on classification tasks, which may only evaluates the retrieval ability of LCLMs. Unlike previous studies, our work provides a more comprehensive analysis of many-shot ICL across a diverse set of tasks and multiple models. We introduce novel metrics to measure retrieval skills and the level of task understanding required for each task. We identify a set of ICL tasks suitable for evaluation and present a refined long-context evaluation benchmark with fine-grained categorization based on required retrieval skills and task understanding.

### 2.3 IN-CONTEXT LEARNING

In-context learning (ICL) enables models to quickly recognize and perform tasks during inference by conditioning on a set of provided demonstrations (Brown et al., 2020). Many previous works have sought to understand the mechanisms behind in-context learning (ICL). Xie et al. (2022) suggests that models implicitly perform Bayesian inference during inference, retrieving relevant skills learned during pretraining. Additionally, Lin & Lee (2024) introduces the concept of a dual operating mode in ICL: task learning and task retrieval. With sufficient demonstrations, models can adapt to unseen tasks learned during pretraining, thereby enhancing performance as the number of demonstrations increases. To explore how many-shot ICL operates, Bertsch et al. (2024) modified the attention patterns by restricting attention among individual examples. Their findings suggest that performance improvements primarily arise from retrieving similar examples rather than comprehending the task. However, their experiment is limited to classification tasks. It may also be biased when comparing full attention and block attention, as block attention allows access to more demonstrations. Our work

| Dataset | Task Category | Avg. Tokens / Shot | Max # of Shots | # of Tasks |
|---|---|---|---|---|
| BANKING77 | Intent Classification | 13.13 | 5386 | 1 |
| GoEmotions | Emotion Classification | 15.85 | 5480 | 1 |
| DialogRE | Relation Classification | 233.27 | 395 | 1 |
| TREC | Question Classification | 11.25 | 6272 | 1 |
| CLINC150 | Intent Classification | 8.95 | 7252 | 1 |
| MATH | Math reasoning | [185.52, 407.90] | [286, 653] | 4 |
| GSM8K | Math reasoning | 55.78 | 784 | 1 |
| BBH | Reasoning | [48.27, 243.01] | [406, 2660] | 4 |
| GPQA | MQ - Science | [183.55, 367.02] | [314, 580] | 1 |
| ARC | MQ - Science | [61.54, 61.54] | [1997, 2301] | 2 |
| XLSUM | New Summarization | 621.32 | 220 | 1 |
| FLORES-200 | Translation | [63.63, 101.74] | [570, 1965] | 3 |

Table 1: Dataset Information. GPT-4o tokenizer is used to calculate # of tokens. Max # of shots is the number of shots can be fitted into the 128k context window. For datasets that have multiple subtasks, we list the range for each value. We have 22 tasks in total.

tries to design better experiments to investigate during many-shot ICL what skill each task mainly requires from LCLMs.

## 3 EXPERIMENT SETTING

To investigate many-shot ICL across various tasks and model sizes, we select 11 models ranging from 3.8B to 123B parameters. Our evaluation includes 12 datasets with 22 subtasks, spanning classification, summarization, reasoning, and translation domains. For each task, we randomly sample 200 data points from the test set, using the full test set if it contains fewer than 200 samples.

For each task, we construct prompts for different context window sizes by incrementally adding new demonstrations from the training set to the prompt of the shorter context window size and duplicate training examples if they are insufficient to fill the context window. To ensure a fair comparison, we randomize the order of demonstrations and consistently use the same set of examples across all context sizes. For simplicity, we apply greedy decoding across all models and conduct each experiment using three different random seeds. For the prompt construction, we only include demonstrations and provide minimal task instruction.

### 3.1 DATASETS

We include five datasets for **classification** tasks: BANKING77, GoEmotions, DialogRE, TREC, and CLINC150. For the **summarization** task, we use XLSUM, and for **translation**, we use FLORES-200. Additionally, we incorporate four datasets for **reasoning** tasks: MATH, BBH, and GPQA, and ARC. More details about each dataset can be found in Table 1 and A.

For the MATH, BBH, GPQA, and ARC tasks, we use accuracy as the evaluation metric. Macro F1-score is employed as the metric for all classification tasks. Rouge-L (Lin, 2004) is used for the XLSUM summarization task. ChrF (Popović, 2015) is applied for translation evaluation.

### 3.2 MODELS

The list of models we use in our experiment is: Llama-3.1 8B and 70B (Dubey et al., 2024), GLM-4-9B-Chat (GLM et al., 2024), Mistral Nemo (12B) and Large (123B) (Mistral AI, 2024), Qwen2 7B and 72B (Yang et al., 2024), Phi-3 mini (3.8B), small(7B), and medium(14B) (Abdin et al., 2024), and Jamba 1.5 Mini (12B/52B)(Team et al., 2024c), and Gemini-1.5-Pro (Team et al., 2024b).We only run Gemini-1-5-Pro on our benchmark. We use the instruction-tuned version of all the models. For models with more than 50B, we run the quantized version of the models, and in C, we show that the quantized version exhibits the same trend as the unquantized version with increasing context length.

## 4 WHICH TASKS BENEFIT FROM MORE EXAMPLES?

In this section, we explore the extent to which many-shot ICL enhances model performance across different task types. Previous work has either focused on only classification tasks (Bertsch et al., 2024) or studied only one specific model (Agarwal et al., 2024). In contrast, our analysis provides a comprehensive evaluation of many-shot ICL across both classification and generation tasks using ten open-weights LCLMs, excluding Mistral-Large in this section. We collect tasks from previous work (Bertsch et al., 2024; Agarwal et al., 2024; Li et al., 2024), categorize them into six types: classification, translation, summarization, math reasoning, science reasoning, and symbolic reasoning.[2] The results, illustrated in Figure 1, include aggregated model performance across task types and the correlation coefficients between context lengths and performance from 1k to 64k. We also plot models' performance on individual task in D.

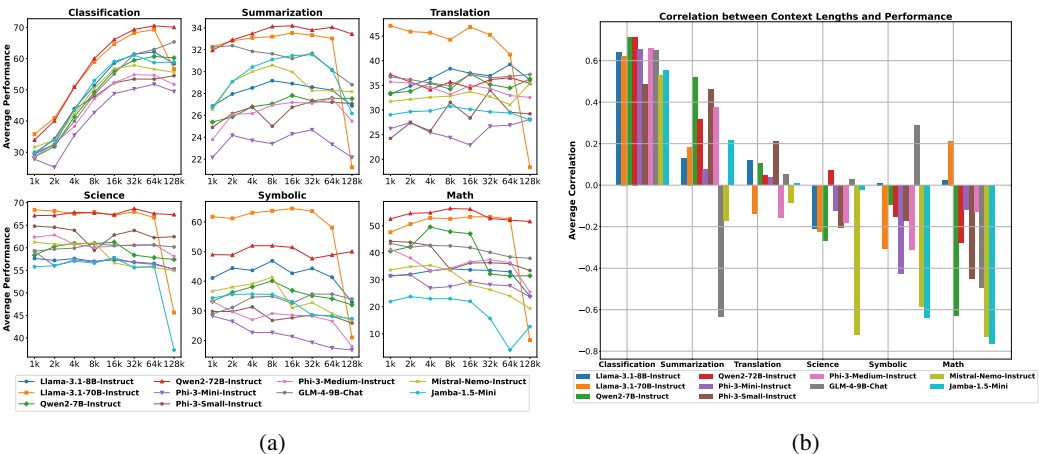

(a)                                                                                      (b)

Figure 1: (a) Aggregated performance of models over datasets in different categories of tasks. (b) Average pearson correlation coefficient between context lengths (1k to 64k) and the corresponding performance.

**Classification performance steadily improves with more shots**: Figure 1a demonstrates a consistent performance increase across all models as more demonstrations are added for classification tasks. This trend indicates a strong positive correlation between context length and performance, which is illustrated in Fig 1b. Given that classification tasks often involve extensive label spaces, e.g., CLINC150 has 150 classes, additional demonstrations provide models with exposure to more classes and thus enhance their ability to perform accurately. This is consistent with prior research findings (Bertsch et al., 2024).

**Subjective tasks do not benefit from more examples**: The GoEmotions task, though being a classification problem, exhibits a fluctuating performance trend across all models with increasing shots in Figure 6. We attribute this inconsistency to the subjective nature of the task, where nuanced emotional categories may lead to low annotator agreement (Demszky et al., 2020). This variance in the annotated labels may results in a weaker correlation between context length and performance. This finding highlights a limitation in using ICL tasks with ambiguous ground truths to evaluate LCLMs, as their performance does not improve with more demonstrations.

**Summarization shows gradual performance gains only**: On summarization, most models exhibit a high correlation between context length and performance. However, there is a noticeable slowdown in the performance gains as the number of demonstrations increases. This suggests that while additional context may improve performance, it does so at a diminishing rate, particularly for smaller models like Llama-3.1-8B that struggle to leverage longer contexts effectively.

**Models' performance fluctuates on translation tasks**: As shown in Figure 7, the performance curves for all models across different languages differ. For the low-resource language, models show larger performance gap than those in the high-resource language, e.g., Spanish. In Chinese, mod-

---

[2]We exclude datasets that are noisy or not open access.

els become spikier than in other languages across different context sizes. In Figure 1a, translation tasks show a very flat curve, with no significant improvement as the number of demonstrations increases. This result contrasts with Agarwal et al. (2024), where the Gemini-1.5 Pro model demonstrated consistent performance improvements in Kurdish and Tamil translation tasks as the context size increased. We think the performance inconsistency is caused by the mismatched multilingual capability of models and different model sizes.

**Math tasks benefit from additional demonstrations, particularly for stronger models**: In math reasoning tasks, only the Llama-3.1 and Qwen2 model families show significant performance improvements with additional demonstrations. Notably, Qwen2 performance plateaus at 16k length, while Llama-3.1 continues to improve until 64k. The models with larger parameter sizes tend to exhibit more consistent performance gains, supporting findings from Agarwal et al. (2024) who have demonstrated that Gemini 1.5 Pro improves on math tasks with more examples.

**Inconsistent trends in science and symbolic tasks**: For science and symbolic reasoning tasks, the performance trends are less predictable, with some models displaying minimal changes when seeing additional examples, while others benefit. *This variability suggests that not all tasks lend themselves to the advantages of many-shot ICL equally*.

Ideally, for every task, additional demonstrations should either improve performance or, at the very least, not harm it. A model with robust long-context capabilities should exhibit a non-decreasing performance trend as the context length increases. Given the inconsistent performance on non-classification tasks and even decreasing performance on some reasoning tasks, in the next two sections, we further investigate what aspects these datasets evaluate and identify a set of tasks useful for evaluating important skills of LCMLs.

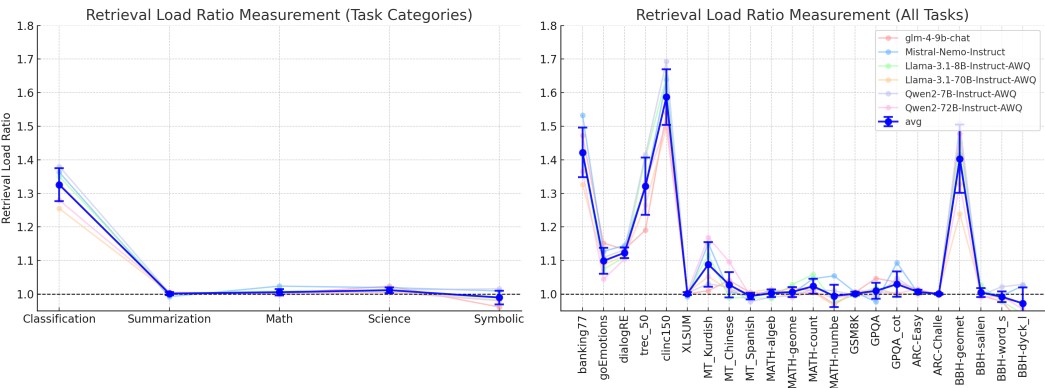

Figure 2: Retrieval Load Ratio on different categories of tasks from 1k to 64k tokens. The ratio of 1 indicates models are not doing retrieval during ICL. Classification is the only category of tasks that has a very high ratio, which means classification tasks requires models retrieval skill during ICL. The rest of tasks is close to 1, and models' performance on these tasks do not rely on retrieving similar examples.

## 5 TASK CATEGORIZATION: RETRIEVAL VS. GLOBAL CONTEXT UNDERSTANDING

To understand what skill each ICL task primarily requires from LCLMs, in this section, we first measure the **retrieval load** of each task and divide them into *retrieval* vs. *non-retrieval* tasks (5.1). Among non-retrieval tasks, we then conduct experiments to identify tasks that truly benefit from additional demonstrations and measure the model's global context understanding skill.(5.2)

### 5.1 RETRIEVAL TASKS

To identify retrieval tasks, we propose a simple metric, **retrieval load ratio**, to assess whether tasks predominantly rely on models to retrieve relevant examples during many-shot ICL. We consider

retrieval load as the retrieval skill required by LCLMs to solve a ICL task. Concretely, for each ICL task, we create two variants of the original demonstrations at each context size ranging from 1k to 64k by removing the 10% most similar and the 10% least similar examples. The model's performance on these variants is then evaluated, and we have $score_{most}$ for removing similar examples and $score_{least}$ for removing dissimilar examples. Here we use BM25 retriever to calculate the similarity. We then average the ratios between $score_{least}$ and $score_{most}$ from 1k to 64k lengths as:

$$\text{Retrieval Load Ratio} = \frac{1}{7} \sum_{l=1k}^{64k} (\frac{score_{least}}{score_{most}})_l \tag{1}$$

Intuitively, if a model predominantly relies on retrieval for a task, removing most similar examples will result in a more pronounced performance drop compared to removing dissimilar ones, which causes the ratio to be larger than 1. Conversely, if there is minimal difference between the two, it means the model does not retrieve similar examples to perform the task, and the ratio will be close to 1.

**Classification tasks requires high retrieval load:** As shown in Figure 2, *all classification tasks exhibit high retrieval load ratio across the six models*. The BBH geometric shapes task also shows a high retrieval raio, indicating that tasks like BANKING77, CLINC150, and TREC50 demand strong retrieval capabilities from the models. Tasks such as GoEmotions and dialogRE have relatively lower retrieval ratios, suggesting they require moderate retrieval skills. Among the symbolic tasks, BBH-geometric_shapes is the only reasoning task that has a high retrieval load ratio. This task involves determining the geometric shape given a full SVG path element, making it similar to a classification task. The high retrieval load ratio of classification tasks can possibly explain the largest positive correlation between performance and context lengths, as displayed in Fig 1b.

**Tasks with low retrieval load**: All the non-classification tasks have a low retrieval load ratio. In Figure 1, models show inconsistent correlations on performance and context lengths for different non-retrieval tasks. This inconsistency may be attributed to the incapability of the LCLMs or the nature of the tasks, which we will investigate more in the next section.

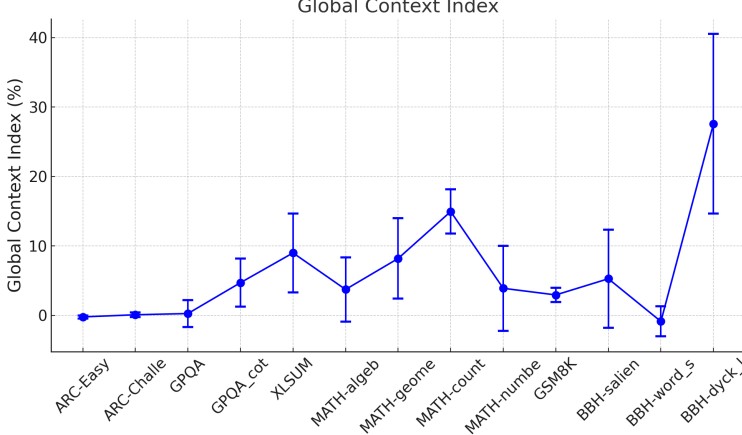

Figure 3: Global context index is the average % difference between adding duplicated vs. unique examples from 2k to 16k context for non-retrieval tasks. 0% means duplicating does not harm the model's performance. Easy tasks such as ARC and word sorting do not benefit from additional information. When a task is too difficult, e.g., GPQA, the model cannot effectively learn all demonstrations unless explanations are provided.

## 5.2 GLOBAL CONTEXT UNDERSTANDING TASKS

In this section, we investigate whether non-retrieval tasks truly benefit from additional demonstrations and whether models use all the demonstrations to understand the task during ICL. We exclude

the translation tasks from this set of experiments due to inconsistent tokenization for different languages and mismatched multilingual capability of models.

**Global Context Index**: We propose another metric, **global context index**, to measure the global context understanding skill required by a task. Specifically, for each non-retrieval task, we have two variants of demonstrations, which both start with the same demonstrations used in the 1k context-length experiment. From 2k to 16k, the unique variant will keep adding unique demonstrations to the prompt, whereas the duplicate variants will repeat the same demonstrations in the 1k length. We denote the performance of the unique variant $score_{unique}$ and the performance of duplicate variant $score_{duplicate}$. Then, we average the percentage difference between $score_{unique}$ and $score_{duplicate}$ from 2k to 16k lengths as:

$$\text{Global Context Index} = \frac{1}{4} \sum_{l=2k}^{16k} (\frac{score_{unique} - score_{duplicate}}{score_{unique}})_l \tag{2}$$

If duplicating examples results in worse performance on a non-retrieval task than adding unique examples, the global context index will be positive and suggests that the model benefits more from unique demonstrations. This means that performance improvements come from learning from diverse examples rather than simply picking up on formatting patterns or relying on spurious correlations between in-domain tokens and predictions. Since non-retrieval tasks typically do not rely on retrieving similar examples, we can conclude that the performance gain on these tasks is likely due to the models' improved global context understanding when more demonstrations are available.

We use Llama-3.1-70B for the experiment because it is best at using additional demonstrations out of all models we have tested so far, e.g., it shows a high positive correlation between context lengths and performance in Fig 1b. Then, we only conduct the experiment up to 16k to minimize the impact of the model's long context capability.

**Global context understanding tasks**: In Figure 3, tasks such as the math problems and summarization, Dyck languages, translation error detection from BBH, and GPQA with explanations all have worse performance with duplicated demonstrations. This means that *they necessitate a greater degree of global context understanding rather than relying on the retrieval of relevant examples*. These tasks are often complex reasoning challenges, for which models may lack pretraining skills to solve perfectly, underscoring the need for additional demonstrations or deeper task comprehension.

**ICL Tasks that are not suitable for LCLM evaluation**: In Figure 3, ARC-Easy, ARC-Challenge, GPQA, the BBH word sorting tasks are indifferent to duplicating examples. This indicates that these tasks do not benefit from additional demonstrations. Most of these tasks assess the intrinsic abilities of the models reasoning with their parametric knowledge, thus a few demonstrations suffice. Adding more demonstrations may introduce distractions rather than improve performance. Interestingly, GPQA with "chain-of-thoughts" benefit from additional examples. We suspect that without these solution steps, GPQA is too challenging for the model to understand even after seeing many demonstrations with answers only.

## 6 MANYICLBENCH: A MANY-SHOT ICL BENCHMARK TO MEASURE RETRIEVAL SKILL AND GLOBAL CONTEXT UNDERSTANDING

In this section, we present a new long-context benchmark MANYICLBENCH, designed to evaluate LCLMs' retrieval skills and global context understanding capabilities using the ICL setup. Based on the results from Section 5, we group tasks into two types:

• **5 Retrieval Tasks**: BANKING77, dialogRE, TREC50, CLINC150, and the geometric shape task from BBH.

• **9 Global Context Understanding Tasks**: all math tasks, summarization task, GPQA with explanations, translation error detection, and dyck language task from BBH.

Evaluation results of popular LCLMs are summarized in Table 2.

**Most models struggle at retrieving examples after 32k length**: Up to a context length of 16k, *all models demonstrate a steady performance increase, indicating effective retrieval from shorter contexts*. However, performance begins to decline after reaching 32k tokens, particularly for the Mistral

| Retrieval Tasks | 1k | 2k | 4k | 8k | 16k | 32k | 64k | 128k | AVG. | AVG.L. |
|---|---|---|---|---|---|---|---|---|---|---|
| GLM-4-9b-Chat | 31.63 | 34.99 | 46.37 | 57.27 | 63.61 | 68.34 | 72.16 | 72.93 | 55.91 | 71.14 |
| Mistral-Nemo-Instruct | 33.44 | 35.45 | 48.17 | 57.95 | 65.38 | 65.49 | 63.61 | 61.73 | 53.90 | 63.61 |
| Mistral-Large-Instruct-AWQ | 49.15 | 51.23 | 60.78 | 71.95 | 77.10 | 79.45 | 77.77 | 61.89 | 66.16 | 73.04 |
| Llama-3.1-8B-Instruct-AWQ | 32.13 | 34.63 | 45.76 | 57.39 | 66.18 | 70.02 | 70.55 | 65.85 | 55.31 | 68.81 |
| Llama-3.1-70B-Instruct-AWQ | 38.75 | 42.87 | 53.98 | 66.07 | 73.12 | 76.56 | 78.48 | 65.56 | 61.92 | 73.53 |
| Qwen2-7B-Instruct-AWQ | 30.18 | 34.03 | 44.40 | 54.85 | 62.92 | 65.91 | 66.94 | 66.38 | 53.20 | 66.41 |
| Qwen2-72B-Instruct-AWQ | 36.41 | 41.89 | 54.24 | 65.33 | 73.39 | 76.53 | 77.51 | 77.47 | 62.85 | 77.17 |
| Phi-3-Mini-Instruct | 30.27 | 30.90 | 38.09 | 48.14 | 53.58 | 57.29 | 56.83 | 48.72 | 45.48 | 54.28 |
| Phi-3-Medium-Instruct | 31.73 | 33.55 | 39.10 | 49.83 | 58.29 | 61.17 | 60.63 | 45.32 | 47.45 | 55.70 |
| Phi-3-Small-Instruct | 31.48 | 36.27 | 46.20 | 54.34 | 59.63 | 59.73 | 60.20 | 48.97 | 49.60 | 56.30 |
| Jamba-1.5-Mini | 32.10 | 36.91 | 48.61 | 60.29 | 66.05 | 68.33 | 66.02 | 65.17 | 55.44 | 66.51 |
| Gemini-1.5-Pro | 36.40 | 47.31 | 58.01 | 65.49 | 71.43 | 74.22 | 72.43 | 72.42 | 62.21 | 73.03 |
| **Global Context Understanding Tasks** | **1k** | **2k** | **4k** | **8k** | **16k** | **32k** | **64k** | **128k** | **AVG.** | **AVG.L.** |
| GLM-4-9b-Chat | 36.79 | 36.23 | 38.30 | 39.30 | 37.60 | 37.94 | 36.53 | 35.45 | 37.27 | 36.64 |
| Mistral-Nemo-Instruct | 33.94 | 34.88 | 34.92 | 34.72 | 28.22 | 28.64 | 26.28 | 23.23 | 30.60 | 26.05 |
| Mistral-Large-Instruct-AWQ | 57.09 | 56.30 | 56.21 | 56.12 | 56.43 | 53.33 | 42.98 | 13.10 | 48.94 | 36.47 |
| Llama-3.1-8B-Instruct-AWQ | 31.31 | 32.79 | 33.02 | 34.50 | 34.25 | 35.22 | 33.71 | 27.88 | 32.84 | 32.27 |
| Llama-3.1-70B-Instruct-AWQ | 45.53 | 47.60 | 48.39 | 49.08 | 49.64 | 49.83 | 47.74 | 13.88 | 43.99 | 37.23 |
| Qwen2-7B-Instruct-AWQ | 37.75 | 39.47 | 43.86 | 44.55 | 42.83 | 35.17 | 33.00 | 32.70 | 38.67 | 33.62 |
| Qwen2-72B-Instruct-AWQ | 47.38 | 49.03 | 50.32 | 50.69 | 50.78 | 48.56 | 48.18 | 48.68 | 49.20 | 48.47 |
| Phi-3-Mini-Instruct | 29.86 | 29.20 | 26.61 | 26.95 | 27.65 | 26.34 | 25.54 | 23.08 | 26.90 | 24.98 |
| Phi-3-Medium-Instruct | 37.74 | 37.15 | 31.49 | 32.02 | 33.04 | 33.19 | 33.06 | 24.56 | 32.78 | 30.27 |
| Phi-3-Small-Instruct | 38.40 | 38.40 | 38.35 | 31.69 | 34.04 | 34.59 | 33.74 | 32.46 | 35.21 | 33.60 |
| Jamba-1.5-Mini | 27.86 | 29.04 | 28.93 | 28.86 | 27.86 | 24.92 | 23.12 | 22.42 | 26.63 | 23.48 |
| Gemini-1.5-Pro | 58.26 | 60.88 | 61.30 | 65.20 | 65.05 | 65.12 | 62.38 | 63.61 | 66.20 | 66.92 |

Table 2: Model performance on retrieval and global context understanding tasks. AVG. is the average model performance of all context lengths. AVG.L. is the average model performance of 32k, 64k and 128k. Red indicates performance improvement compared to 1k. Blue indicates performance downgrade compared to 1k. A darker color means higher improvement or downgrade. **BOLD** number means the largest number of a column. Many models start downgrading their performance after 32k on retrieval tasks. On global context understanding tasks, many models start struggling even before 16k.

family and Jamba models. After 64k, the Llama 3.1 family and the mini and medium versions of Phi-3 exhibit a notable downgrade in performance. In contrast, the Qwen-2 family maintains robust performance, with minimal degradation from 64k to 128k. Remarkably, only GLM-4 continues to improve in retrieval performance beyond 64k, indicating its impressive retrieval capabilities within a very long context window. Interestingly, larger models like Mistral-Large and Llama-3.1-70B exhibit the most significant performance losses as context length increases, suggesting that size alone does not ensure superior long-context retrieval ability.

**Challenges in global context understanding tasks**: Global context understanding tasks prove to be more challenging, with *many models struggling even at short context lengths like 2k or 4k*. Only the Llama 3.1 family, Qwen2 family, and GLM-4 models effectively leverage many demonstrations up to 16k. At 32k, only the Llama 3.1 models sustain performance. As context length extends from 32k to 128k, all models experience performance degradation, highlighting that current architectures still struggle to grasp global context and utilize demonstrations effectively. Notably, Qwen2-72B and GLM-4 are the only models that do not experience significant performance drops in this category.

**The paradox of model size**: Despite the common assumption that larger models possess greater capabilities, our findings illustrate that larger models can experience more substantial performance losses compared to smaller models if not trained adequately on long-context data. For instance, Mistral-Large (123B) shows optimal performance from 1k to 32k but experiences a dramatic drop beyond 32k, which is worse than Phi-3-Mini (3.8B). A similar trend is observed with Llama-3.1-70B at 128k. Both underscore the importance of targeted training for long-context tasks.

**Llama 3.1 performance and training limitations**: The Llama 3.1 models initially capitalize on additional demonstrations effectively up to 64k but suffer significant performance declines at 128k. This pattern aligns with trends observed in other long-context evaluation benchmarks (Hsieh et al., 2024). We suspect that these performance drops are linked to insufficient training with long-context data during the supervised fine-tuning (SFT) stage. According to Table 7 in (Dubey et al., 2024), the average token count for long-context datasets is around 38k, indicating limited exposure for models to effectively learn from data points at 128k lengths.

**Qwen2 and GLM-4 show relatively robust capabilities on both tasks**: The Qwen2-72B model consistently maintains performance across both retrieval and global context understanding tasks, demonstrating its adaptability for longer contexts. Trained on data with up to 32k tokens, Qwen2 models employ modified RoPE frequency and training-free positional interpolation methods to handle longer contexts. However, the Qwen2 family models drop their performance from 16k to 32k in the global context of understanding tasks but maintain their performance after 32k. This raises the question of whether the training-free length extension methods enable models to use additional demonstrations or merely maintain their performance in the short context length and ignore additional examples during many-shot ICL. Meanwhile, GLM-4-chat also shows a relatively robust performance at a longer context size and is the only model to experience a performance increase from 64k to 128k on retrieval tasks. GLM-4's training methodology closely mirrors that of Llama 3.1 models, with adjustments to the RoPE base and continuous training on long-context data. The difference is, during SFT, GLM-4-9B follows LongAlign (Bai et al., 2024), which determines the length distribution of the long-context SFT data carefully. GLM-4-9B also goes through the RLHF stage with both short and long data.

**Gemini-1.5-Pro shows a very robust long context capability**: Similar to other open-weight models on retrieval tasks, Gemini-1.5-Pro begins to show performance degradation beyond 32k. However, it is one of only three models (alongside Qwen-2-72B and GLM-Chat-9B) that demonstrate impressive retrieval capabilities beyond 64k and maintain performance at 128k. On global context understanding tasks, Gemini-1.5-Pro significantly outperforms other open-weight models, showcasing its ability to grasp the global context and effectively utilize all the demonstrations.

**Future directions** can be investigating the optimal length distribution of both pre-training and SFT long-context data, as well as studying the effects of continual training on long-context data and the implementation of training-free length extension methods.

# 7 CONCLUSION

We investigated many-shot in-context learning (ICL) across various tasks using different open-weight models, assessing their suitability for evaluating long-context language models (LCLMs). Our findings indicate that classification and summarization tasks consistently benefit from additional demonstrations, while other tasks do not. To identify a set of tasks suitable for long-context evaluation, we introduced the concept of retrieval load ratio to assess the retrieval demands of different tasks. This analysis revealed that classification tasks predominantly rely on the model's retrieval capabilities. For non-retrieval tasks, we conducted duplication experiments to differentiate global context understanding tasks from those that introduce noise. Based on these insights, we categorized tasks into two distinct groups: retrieval tasks and global context understanding tasks. Furthermore, we introduced a novel many-shot ICL benchmark, **ManyICLBench**, designed to evaluate both retrieval and global context understanding skills of LCLMs. Benchmarking open-weight LCLMs on ManyICLBench revealed that most models struggle with global context understanding tasks at lengths below 16k tokens. In contrast, performance on retrieval tasks tends to decline after 32k tokens.

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

## A DATASETS

**BANKING77** (Casanueva et al., 2020) is an intent classification task in the banking domain. It has over 10k customer service queries labeled with 77 intents.

**GoEmotions** (Demszky et al., 2020) contains 58 Reddit comments labeled for 27 emotion categories or Neutral.

**DialogRE** (Yu et al., 2020) is a relation extraction dataset that is built based on transcripts of an American TV show Friends. It comprises 10,168 relation triples for 1,788 dialogues and 36 total relations types. We only focus on relation classification for this dataset.

**TREC** (Li & Roth, 2002; Hovy et al., 2001) is a question classification dataset with six coarse and 50 fine class labels. It contains 5,500 questions in the training set and 500 in the test set.

**CLINC150** (Larson et al., 2019) is an intent classification dataset with 150 intents from 10 domains.

**MATH** (Hendrycks et al., 2021) is a dataset of 12,5000 challenging completion mathematics problems. Each problem has a full step-by-step solution. We use four subdomains from the dataset: algebra, geometry, counting and probability, and number theory.

**GSM8K** (Hendrycks et al., 2021) consists of 8.5K high quality grade school math problems created by human problem writers. These problems take between 2 and 8 steps to solve, and solutions primarily involve performing a sequence of elementary calculations using basic arithmetic operations (+ - / *) to reach the final answer.

**BBH** (Srivastava et al., 2022) is a subset of 23 challenging BIG-Bench tasks (Suzgun et al., 2022), which include task categories such as mathematics, commonsense reasoning, and question answering. We use four subtasks from BBH-Hard: geometric shape, salient translation error detection, word sorting, and dyck languages.

**ARC** (Clark et al., 2018) is a dataset of 7,787 genuine grade-school level, multiple-choice science questions. The dataset is partitioned into a Challenge Set and Easy Set, where the former contains only questions answered incorrectly by both a retrieval-based algorithm and a word co-occurrence algorithm.

**GPQA** (Rein et al., 2023) is a dataset of 448 multiple-choice questions with detailed explanations written by domain experts in biology, physics, and chemistry.

**XLSUM** (Hasan et al., 2021) is a summarization dataset that focuses on news articles from BBC. In this work, we focus only on English news articles.

**FLORES-200** (NLLB Team, 2022) is a translation benchmark that contains many low-resource languages. We follow Agarwal et al. (2024) and choose the translation task from Tamil to English. Additionally, we also test models on Chinese and Spanish.

## B MODELS

**Llama-3.1 8B and 70B** (Dubey et al., 2024): We use both the 8B and 70B Llama 3.1 Instruction models. These multilingual models are trained on a 128k context window using position interpolation. The models are further fine-tuned with synthetic long-text Supervised Fine-Tuning (SFT) data and also undergo Direct Preference Optimization (DPO) (Rafailov et al., 2024).

**GLM-4-9B-Chat** (GLM et al., 2024): This is a 9-billion-parameter multilingual model, also trained on a 128k context window with position interpolation. It is further fine-tuned with labeled long-text SFT data and undergoes a DPO stage.

**Mistral Family** (Mistral AI, 2024): We use both 12-billion-parameter and 123-billion-parameter multilingual models, trained on a 128k context window.

**Qwen2 7B and 72B** (Yang et al., 2024): These two models are trained with a context size of 32k tokens, and their context window is extended to 128k by YARN (Peng et al., 2023), a dynamic position interpolation technique.

**Phi-3** (Abdin et al., 2024): We use the mini (3.8B), small (7B), and medium (14B) versions of Phi-3 models. They are trained with the context size of 4k tokens on high quality data, and LongRope (Ding et al., 2024) extends their context size to 128k.

**Jamba-1.5-Mini** (Team et al., 2024c): It's a hybrid SSM-Transformer model with 12B of active parameters and 52B of total parameters with a context size of 256k tokens.

**Gemini-1.5-Pro**: It is a commercial model introduced by Google and has a context size of 2 million tokens.

## C  QUANTIZATION VS. REGULAR

We compare the 4-bit quantized version and unquantized version of both Llama-3.1 8B and Llama-3.1-70B. In both Figure 4 and Figure 5, we can observe that the quantized version experiences a little performance drop but exhibits the same trend as the unquantized version with the increasing context length.

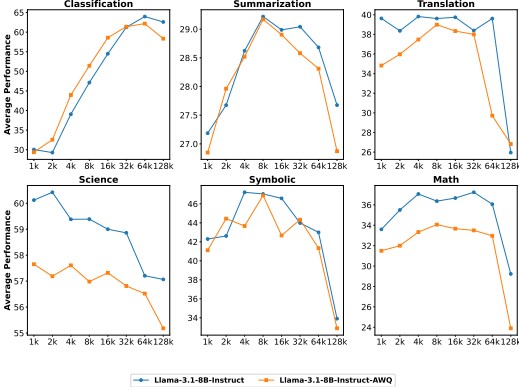

Figure 4: Comparison between Llama-3.1-8B and 4-bit quantized Llama-3.1-8B. There are some performance gaps between two models on translation, science, and math tasks, but with the increasing context size, the performance trend is the same for both models.

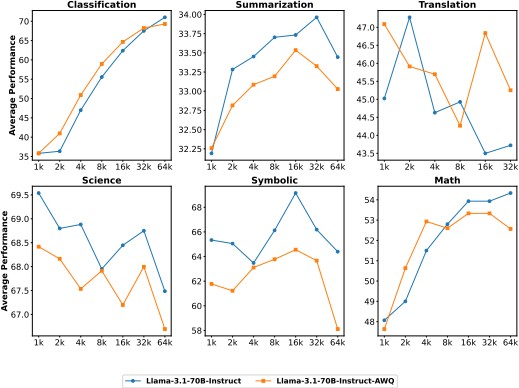

Figure 5: Comparison between Llama-3.1-70B and 4-bit quantized Llama-3.1-70B. Similar to the smaller model, the performance trends hold for both models except the translation tasks. In our benchmark, we exclude all the translation tasks because of the inconsistent multilingual ability of LCLMs.

# D  TASK PERFORMANCE

In this section, we present the models' performance on individual tasks and group them by the task categories: classification, translation, summarization, and reasoning.

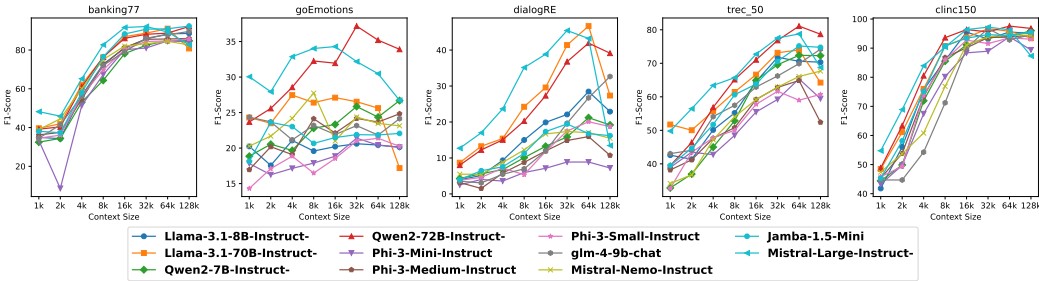

Figure 6: Models' performance on all classification tasks. All tasks except GoEmotions show a very consistent gain with increasing context size. We excluded GoEmotions from our benchmark because of the data's strong subjectivity.

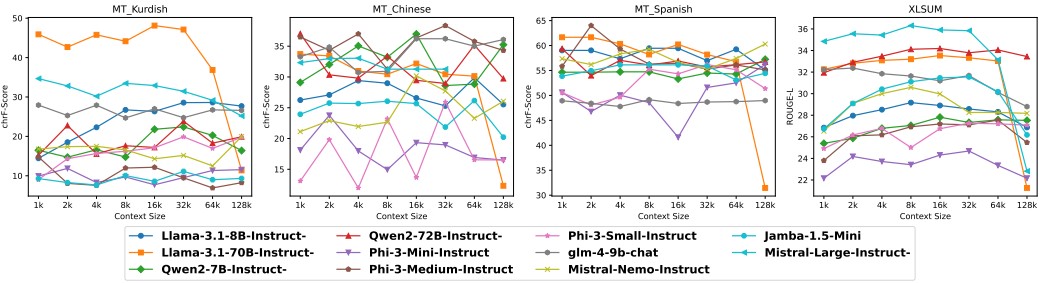

Figure 7: Models' performance on all translation tasks and the summarization task. For translation tasks, we do not observe a clear pattern among different languages and models, which can be caused by LCLMs' different multilingual abilities. We can see a slightly positive trend for the summarization task.

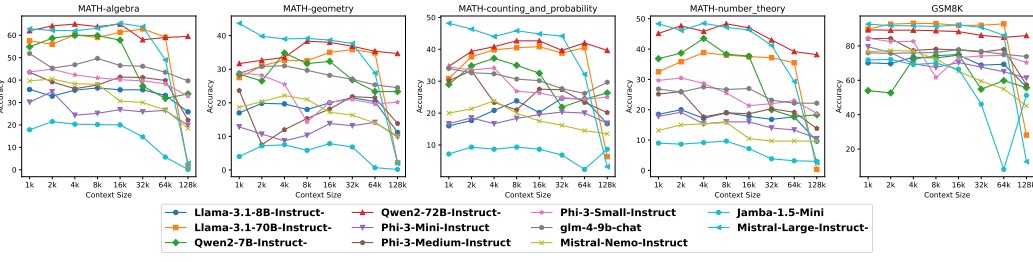

Figure 8: Models' performance on all math tasks. Overall, the larger and stronger models benefit more from the increasing context window size on math tasks.

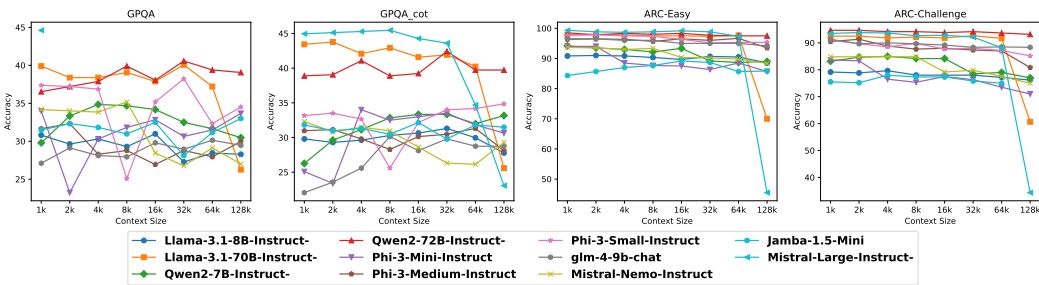

Figure 9: Models' performance on all science tasks. For the ARC task, the performance of all models stays the same across all context sizes. For GPQA, we can see larger and more robust LCLMs keep or increase their performance with the increasing context size.

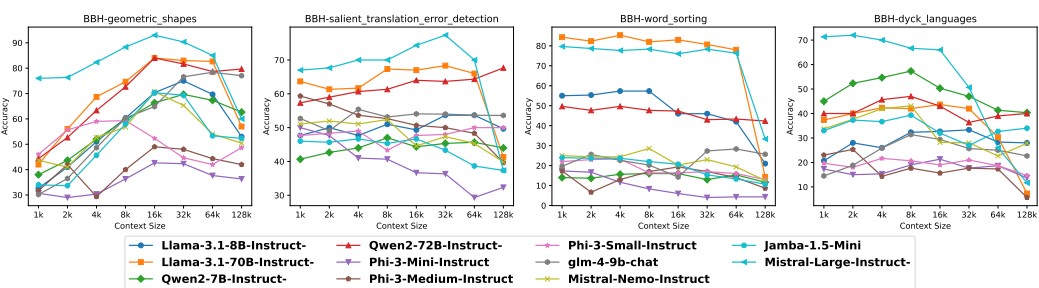

Figure 10: Models' performance on all symbolic tasks. For the geometric shape and translation error detection tasks, we can all model benefit from the increasing context length. We suspect the word sorting task may too easy for the models, so the lines are flat. For the dyck language task, the models experience performance gain up 16k context length but start downgrading afterward.

# E    ADDITIONAL RETRIEVAL LOAD EXPERIMENTS

To ensure the performance downgrade is not caused by the absence of certain labels in the retrieval load experiment from Section 5, we replace similar examples with distant examples with the same labels. The new retrieval load ratio formula is $\frac{score_{original}}{score_{replace}}$. We use Llama-3.1 models and conduct this experiment from 1k to 64k with both BM25 and SBERT (Reimers & Gurevych, 2019) retrievers and exclude XLSUM.

**BM25**: The trend in Figure 2 matches the results of Figure 11. All the classification tasks downgrade performance more when similar examples are replaced. However, the degree of downgrade is less significant than removing similar examples.

**SBERT**: For SentenceTransformer, we use multi-qa-MiniLM-L6-cos-v1 as the base model. In addition to XLSUM, we exclude geometric shape, Dyck language, and dialogRE because the inputs of the first two tasks are mainly symbols and numbers, and the input of dialogRE is too long for the retriever to be effective. The trends observed from Figure 2 and Figure 11 still hold in Figure 12. That is, all the classification tasks still have a higher ratio and the non-classification tasks have a ratio close to 1.

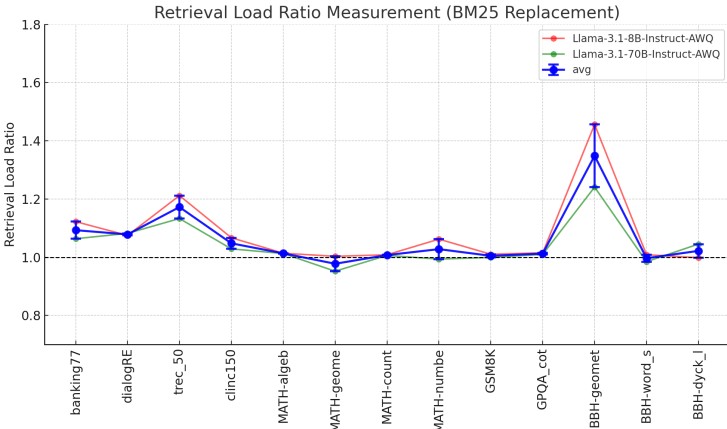

Figure 11: Retrieval Load Ratio under the replacement setting with BM25 on all tasks expect XL-SUM from 1k to 64k tokens. The ratio of 1 indicates models are not doing retrieval during ICL because similar demonstrations don't help models perform better. Similar to Figure 2, classification is the only category of tasks that has a higher ratio, which means classification tasks largely require model retrieval skills during ICL. The rest of the tasks is close to 1, and the models' performance on these tasks does not rely on retrieving similar examples.

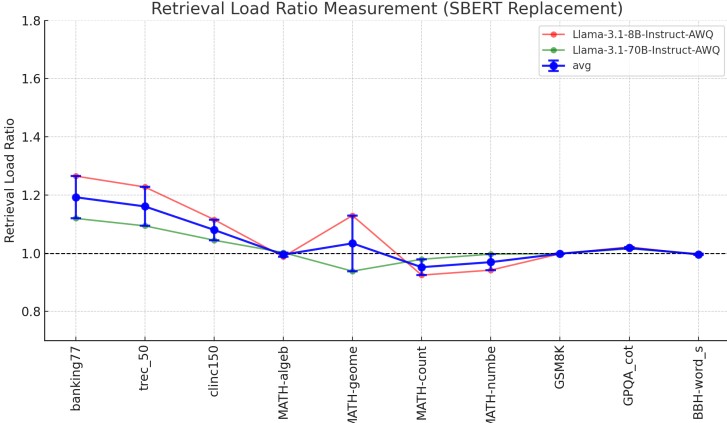

Figure 12: Retrieval Load Ratio under the replacement setting with SBERT on selective tasks from 1k to 64k tokens. A ratio of 1 signifies that models do not perform retrieval during in-context learning (ICL), as similar demonstrations do not enhance their performance. As shown in Figure 2, classification tasks are the only category with a higher retrieval load ratio, indicating a strong dependence on retrieval during ICL. In contrast, other tasks exhibit ratios close to 1, suggesting minimal reliance on retrieval, with models' performance largely unaffected by retrieval-based demonstrations.

# F    ADDITIONAL GLOBAL CONTEXT INDEX RESULT

In Figure 13, We present the global context index for each non-retrieval task with input lengths of up to 64k, observing results consistent with those from the 16k input length setup.

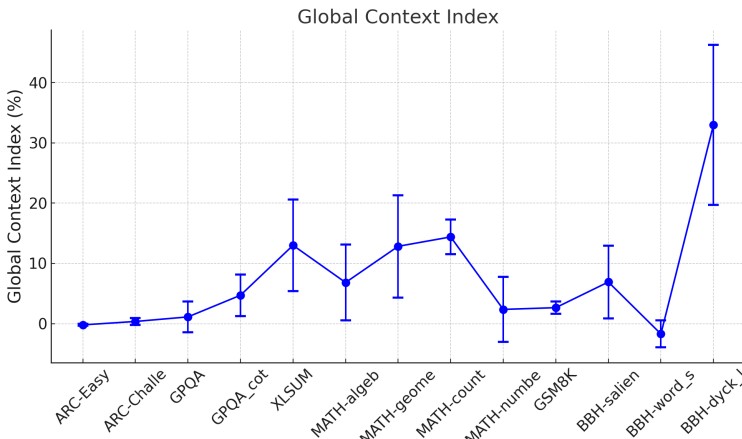

Figure 13: Global context index is the average % difference between adding duplicated vs. unique examples from 2k to 64k context for non-retrieval tasks. 0% means duplicating does not harm the model's performance. Easy tasks such as ARC and word sorting do not benefit from additional information. When a task is challenging, e.g., GPQA, the model cannot effectively learn all demonstrations unless explanations are provided.

