# OpenReview forum: "Retrieval or Global Context Understanding? On Many-Shot In-Context Learning for Long-Context Evaluation"
_ICLR.cc/2025/Conference — ICLR 2025 Conference Withdrawn Submission_

### Official Review · Reviewer_45kP · 2024-10-28

**Soundness:** 2
**Presentation:** 3
**Contribution:** 2
**Rating:** 5
**Confidence:** 3

**Summary:**

This paper explores the capabilities of long-context language models (LCLMs) in handling long-text information, particularly their abilities in global context understanding versus retrieval capabilities. The paper assesses LCLMs through many-shot In-Context Learning (ICL) and introduces a new benchmark, MANYICLBENCH, to separately measure the retrieval skills and global context understanding capabilities of LCLMs.

The key findings and contributions are:
1. Classification and summarization tasks show significant performance improvements with additional demonstrations, while translation and reasoning tasks do not exhibit clear trends.
2. Tasks are categorized into retrieval tasks and global context understanding tasks by analyzing performance changes when removing different examples.
3. A new benchmark, MANYICLBENCH, is introduced to evaluate the retrieval and global context understanding capabilities of LCLMs.

**Strengths:**

1. Originality: This paper pioneers a new evaluation paradigm for LCLMs by differentiating between retrieval and global context understanding skills and also .introduce a new benchmark MANYICLBENCH providing new tools for evaluating long-text models.
2. Quality: The study offers a rigorous experimental design and insightful data analysis, though it could benefit from broader model diversity.
3. Significance: It provides valuable contributions to LCLM evaluation and practical applications, suggesting promising directions for future research.

**Weaknesses:**

1. Task Coverage: While the paper covers a variety of task types, there may still be other types of tasks not fully considered, such as dialogue systems or multi-document summarization.
2. Lack of practicality: At present, exploratory research on LCLMs may change with the iteration of model versions. Some conclusions of the article have certain reference significance for future progress, but not much practical significance.

**Questions:**

1. Does the paper discuss specific challenges encountered in long-text processing, such as information forgetting or context confusion?
2. Do models show a performance decline when processing extremely long texts, and is this related to the model's capacity or training methods?

---

> ### Author Response · Authors · 2024-11-24
>
> Thank you for your valuable feedback. We aim to address your comments below:
>
> > **1. Task Coverage**
>
> In our paper, we currently focus on common in-context learning (ICL) tasks. For the revision, we plan to expand our scope to include additional tasks such as dialogue systems and multi-document summarization. However, our main concern lies in evaluating the outputs of dialogue systems and determining whether such evaluations might be overly challenging. If you have any suggestions regarding suitable datasets or evaluation metrics, we would greatly appreciate your input.
>
> > **2. Lack of practicality**
>
> Thank you for recognizing the significance of our work in the strengths section. In this paper, we identify suitable in-context learning (ICL) tasks for evaluating long-context language models (LCLMs), characterize the primary skills measured by each task, and propose MANYICLBENCH—a benchmark designed to evaluate models’ retrieval capabilities and global context understanding.
>
> Our work enhances the current understanding of many-shot ICL for LCLMs and provides valuable insights into which tasks benefit from many-shot demonstrations. Additionally, we develop methods to characterize each task, offering future researchers a systematic way to categorize tasks and datasets.
>
> Most long-context applications, such as multi-document summarization and code repository understanding, require models to effectively combine retrieval skills with global context understanding. We believe MANYICLBENCH will serve as a valuable tool for evaluating long-context models. Furthermore, the analyses we perform on the benchmark shed light on the current progress of LCLMs and provide guidance for developing more effective long-context training algorithms in the future.
>
> >**Q1**
>
> Since our work primarily focuses on evaluating models' retrieval and global context understanding skills, it serves as a complementary approach to studying information forgetting and context confusion. We can examine how the position of demonstrations impacts the extent of information forgetting by altering their placement within the prompt. Additionally, by including more demonstrations, we stress-test LCLMs in long-context scenarios, investigating whether models become confused when faced with an increased number of contexts or demonstrations.
>
> In future work, we plan to delve deeper into these specific challenges, providing further insights into the limitations and strengths of LCLMs in handling long-context tasks.
>
> >**Q2**
>
>  In Table 2, almost all models experience a performance downgrade when extending the context length from 64k to 128k. This degradation is likely tied to the context length extension methods used during model training. An effective training algorithm should ensure that the model maintains its performance across extended contexts.
>
> For instance, Llama-3.1-8B demonstrates stronger retrieval capabilities than Qwen2-7B at context lengths of 64k or smaller. However, when the context length increases to 128k, Llama-3.1-8B suffers a 7% performance downgrade, whereas Qwen2-7B maintains its performance. At the end of Section 6, we discuss how Qwen2 models might employ a more effective approach to extending context size compared to Llama-3.1 models.
>
>
> We hope we have addressed your concerns and kindly ask you to re-evaluate your assessment. Please feel free to reach out if you have any additional questions.

---

> > ### Comment · Reviewer_45kP · 2024-11-25
> > **Thank you for responding to my concern.**
> >
> > Thank you for responding to my concern, but I feel that there is still a slight lack of innovation. I plan to keep my score unchanged.

---

> > > ### Author Response · Authors · 2024-12-02
> > >
> > > Thank you for your reply. We want to emphasize further the innovation and contributions we make.
> > >
> > > * Investigate whether ICL tasks benefit from additional demonstrations and assess their suitability for evaluating LCLMs with a context length up to 128k tokens.
> > >
> > > * Develop methods to characterize the primary skills evaluated by in-context learning (ICL) tasks, with a particular focus on distinguishing between retrieval capabilities and global context understanding. To the best of our knowledge, this is the first work to systematically address this distinction.
> > >
> > > * Construct a many-shot ICL benchmark, named MANYICLBENCH, designed for evaluating LCLMs on both retrieval and global context understanding, while excluding irrelevant
> > > datasets previously used in LCLM evaluation.
> > >
> > > * Benchmark 11 widely-used state-of-the-art LCLMs on MANYICLBENCH to assess their
> > > performance comprehensively

---

### Official Review · Reviewer_pTsV · 2024-11-04

**Soundness:** 2
**Presentation:** 3
**Contribution:** 2
**Rating:** 3
**Confidence:** 4

**Summary:**

This paper aims at exploring whether LLMs rely more on retrieval or global context understanding to perform tasks like text classification, summarization, reasoning, and translation in many-shot ICL with long-context settings. Using BM25 to measure similarity between demonstrations, the authors assume that a performance drop from removing similar examples indicates high reliance on retrieval. Author(s) suggest that classification and summarization tasks need retrieval ability, while reasoning tasks need global context understanding, though the latter concept is not clearly defined in the paper. Extensive experiments were conducted across a collection of 11 tasks.

**Strengths:**

1. a comprehensive analysis on many-shot ICL across a wide range of tasks and long-context LLMs.

2. The problem studied in this paper is interesting, hard to investigate, and may provide insights into a deeper understanding of LLMs.

**Weaknesses:**

1. The paper centers on the distinction between retrieval ability and global context understanding. Yet, it lacks formal definition of these two concepts, and provides limitted discussion on their meanings and implications for LLMs or downstream tasks. I think there is significant overlap between these two concepts.


2. Regarding RQ2 ("What skill does each task primarily measure?"), author(s) state that "a more pronounced drop in performance upon removing similar examples, which indicates the task’s heavy reliance on retrieval capabilities." This statement requires supporting evidence and rationale, as almost all NLP tasks require global context understanding. It may be insufficient to draw this conclusion simply based on similarity between demonstrations and performance drop.


3. The similarity between demonstrations, estimated using BM25, is based on lexical overlap, which is a limited metric. Lexical similarity does not necessarily capture true semantic similarity, nor does it reliably indicate a preference for retrieval or global context understanding.


4. Prior works have demonstrated that lexical overlap as a similarity measure may lead to spurious correlations. This raises questions about the generalizability of the results.


5. Statements such as "classification tasks benefit from more demonstrations" or "classification tasks predominantly test models’ retrieval skills" lack roust supporting evidence. It is not clear why more demonstrations means better retrieval skill. It may be possible that more similar demonstrations can better illustrate the decision boundary.


6. Previous works have shown demonstration order can impact the performance, but this aspect has not been discussed in this work.


Reference:

[1] Calibrate Before Use: Improving Few-Shot Performance of Language Models, Zhao et al., ICML 2021

[2] Fantastically Ordered Prompts and Where to Find Them: Overcoming Few-Shot Prompt Order Sensitivity, Lu et al., ACL 2022

**Questions:**

1. When measuring the similarity of demonstrations for (x, y), do you only consider x or both x and y?

---

> ### Author Response · Authors · 2024-11-24
>
> Thank you for your valuable feedback. We aim to address your comments below:
>
> > **1. Definition of the retrieval ability and global context understanding**
>
> We define retrieval ability as “the model’s capability to pinpoint relevant and similar examples.” While it is true that almost all NLP tasks require some level of global context understanding, even with a single example in the prompt, our focus is on many-shot in-context learning (ICL). In this context, we define global context understanding as “the model’s ability to comprehend multiple examples across the full input.”
>
> In ICL, the model utilizes its retrieval ability to identify examples similar to the test input and make inferences based on them. Simultaneously, it leverages its global context understanding to process and integrate all the demonstrations in the input. Both skills are crucial for achieving strong performance on downstream tasks such as long-document question answering (QA), multi-document summarization, multi-turn dialogue, and long-context retrieval-augmented generation (RAG).
>
> However, many existing studies primarily evaluate a model’s retrieval ability. Our work offers an alternative approach to assess both retrieval and global context understanding skills, providing a more comprehensive evaluation of model performance.
>
> > **2. Regarding Q2**
>
> We believe that our statement is well-supported by the experimental results in Section 5 and the findings of [1]. Their study compares performance between using random samples and similar samples, demonstrating that sampling similar demonstrations yields better performance than sampling random ones.
>
> While we do not claim that retrieval tasks lack a need for global context understanding, we suspect that these tasks primarily rely on a model’s retrieval skills to achieve strong performance. To further support this assertion, we conducted an additional experiment where we replaced the top k% most similar examples with the most distant examples that share the same labels. This ensures that the label distribution remains unchanged, but the examples themselves differ. Even with an identical set of labels, the model exhibited performance loss on classification tasks, indicating its reliance on retrieving similar examples.
>
> Additional details about this experiment can be found in the **Classification Tasks Relies on Retrieval** section of the response and Appendix E.
>
> > **3/4. Using SenteneTransformer as a retriever**
>
> We use BM25 because it is one of the most widely used retrievers. To address your question about the generalizability of the results, in Section 5, we plot the results for Qwen2, Llama-3.1, Mistral-Nemo, and GLM, and compute their average. All six models exhibit similar trends. Additionally, we conducted further experiments where we replaced similar examples with distant examples that share the same labels. Using both BM25 and SentenceTransformer (SBERT), we observed consistent results: the replacement has a greater impact on classification tasks.
>
> Additional details about this experiment can be found in the **Classification Tasks Relies on Retrieval** section of the response and Appendix E.

---

> ### Author Response · Authors · 2024-11-24
>
> > **5. Classification Tasks Relies on Retrieval**
>
> Classification tasks benefit more from demonstrations, as evidenced by our experiment in Section 4. In the classification plot of Figure 1(a), all models exhibit a strong correlation between performance and context length. Prompts with more demonstrations are likely to include more similar examples. This observation is further supported by the retrieval load experiment in Section 5 and the new experiments in Appendix E.
>
> The first experiment shows that removing similar examples hurts the model’s performance more than removing distant examples, indicating that the model relies on similar examples to make predictions. In [2], it is shown that the decision boundary is influenced by the label distribution of the demonstrations. However, the results of the second experiment demonstrate that even when the label distribution remains the same—where the decision boundary is minimally impacted—replacing similar examples with distant examples still degrades performance. This indicates that the model retrieves similar demonstrations to make predictions, further supporting our statement that "classification tasks predominantly test models’ retrieval skills."
>
> The new experiment results are presented in Figure 12 and Figure 13 from Appendix E, along with the table below:
>
> **Retrieval Load Ratio Results for Original and Replacement Experiments with BM25 and SBERT**
>
>
> | Task                                    |   Remove (BM25) |   Replace (BM25) |   Replace (SBERT) |
> |:----------------------------------------|----------------:|----------------:|------------------------:|
> | banking77                               |        1.37432  |        1.09297  |                1.1924   |
> | dialogRE                                |        1.11089  |        1.07847  |                - |
> | trec_50                                 |        1.32233  |        1.1724   |                1.16085  |
> | clinc150                                |        1.53891  |        1.04783  |                1.08026  |
> | MATH-algebra                            |        1.00061  |        1.01333  |                0.995086 |
> | MATH-geometry                           |        1.01391  |        0.977575 |                1.03627  |
> | MATH-counting_and_probability           |        1.02319  |        1.00746  |                0.952077 |
> | MATH-number_theory                      |        0.981024 |        1.0278   |                0.965633 |
> | GSM8K                                   |        0.997295 |        1.00486  |                0.998426 |
> | BBH-geometric_shapes                    |        1.34485  |        1.34859  |                -  |
> | BBH-salient_translation_error_detection |        1.01839  |        1.00197  |                - |
> | BBH-word_sorting                        |        0.982197 |        0.996679 |                0.996264 |
> | BBH-dyck_languages                      |        0.957303 |        1.02183  |                - |
>
> We excluded XLSUM because computing similarity among many news articles is computationally expensive. For SBERT, we excluded geometric_shapes, Dyck_languages, and dialogRE because the inputs of the first two tasks primarily consist of symbols and numbers, while the inputs for dialogRE are too lengthy for the retriever to be effective.
>
> The results from the original experiment still hold under both retrievers. Classification tasks exhibit a higher retrieval load ratio, while non-classification tasks have ratios closer to 1, suggesting that classification tasks rely more heavily on retrieving similar examples.
>
>
> > **6. Ordering Effect**
>
> As you mentioned, the ordering effect has been well-studied in previous works [1, 3, 4]. We are aware that the order of demonstrations can influence a model’s performance. However, our primary focus is to evaluate the capabilities of long-context models. To minimize the impact of ordering, we shuffle the order of demonstrations across each random seed.
>
> > **Measure similarity**
>
> We only consider x because using y to retrieve examples would make the task overly simple and impractical. The objective of the task is to predict y given x, so initially, we have no prior knowledge of y.
>
> We hope we have addressed your concerns and kindly ask you to re-evaluate your assessment. Please feel free to reach out if you have any additional questions.
>
> [1] Bertsch et al., In-Context Learning with Long-Context Models: An In-Depth
> Exploration, 2024
>
> [2] Zhao et al., Calibrate Before Use: Improving Few-Shot Performance of Language Models, ICML 2021
>
> [2] Lu et al., Fantastically Ordered Prompts and Where to Find Them: Overcoming Few-Shot Prompt Order Sensitivity, ACL 2022
>
> [4] Li et al., Long-context LLMs Struggle with Long In-context Learning, 2024

---

> > ### Comment · Reviewer_pTsV · 2024-11-27
> >
> > I thank authors for their responses. Some points have been addressed but others remain unresolved. I'm uncertain whether retrieval--defined as the model’s capability to pinpoint relevant and similar examples--and global context understanding--defined as the model’s ability to comprehend multiple examples across the full input--are meaningfully distinct. I think one is the subset of the other. I encourage authors to clarify the distinction, address the uncertainty in problem formulation, and analyze the implications of ordering demonstrations or rationalize why random ordering does or does not affect the results. In addition, I suggest providing further analysis of the claims made in the paper, e.g. why adding more demonstrations means better retrieval skill or why classification tasks predominantly test models’ retrieval skills.
> >
> > Regarding the following response:
> >
> > >> even when the label distribution remains the same—where the decision boundary is minimally impacted—replacing similar examples with distant examples still degrades performance
> >
> > decision boundary depends on the "content" of examples, not merely label distribution.

---

> ### Author Response · Authors · 2024-12-02
>
> Thank you for your engagement and for taking the time to delve into our work. We want to clarify some potential confusion you may have and provide further context to ensure our findings and claims are fully understood.
>
> > **Retrieval and Global Context Understanding**
>
> Retrieval can be considered a subset of global context understanding because a model needs to recognize relevant parts of the input to grasp the entire input sequence. However, retrieval and global context understanding are meaningfully distinct. Understanding a small part of the input does not necessarily equate to comprehending the whole sequence.
>
> For example, imagine an input containing 100 unique numbers ranging from 1 to 100. A retrieval task might ask the model to identify the first number greater than 55. In contrast, a global context understanding task could ask for the sum of all numbers greater than 55. While solving the retrieval task involves pinpointing a specific number, solving the global context task requires aggregating all relevant numbers after 55. Success in the retrieval task doesn’t imply the model can perform the more complex aggregation needed for the global context task.
>
> Similarly, in in-context learning (ICL), a model might retrieve a single example that closely matches the query, demonstrating minimal understanding of other examples in the input. However, to solve global context understanding tasks, the model must process and integrate information from all examples, synthesizing them to produce a comprehensive result.
>
> > **Demostration Ordering**
>
> We conducted an additional experiment to investigate how demonstration ordering affects model performance using Llama-3.1-8B. Specifically, we examined the impact of placing the top 10% most similar examples at different positions within the prompt: the front, middle, or end. We then compared the average percent change in performance relative to the original order across input lengths ranging from 1k to 64k.
>
> | Task                                    |   Front   |   Middle   |   Back    |
> |:----------------------------------------|----------------:|----------------:|------------------------:|
> | banking77                               |  -3.56099  |  -1.01339  |   0.424146 |
> | dialogRE                                |  -        |  -        |  -        |
> | trec_50                                 |  -8.81604  |  -4.59433  |   3.89025  |
> | clinc150                                |  -1.84406  |  -1.10688  |   0.536058 |
> | MATH-algebra                            |  -0.4112   |  -0.486733 |   1.33386  |
> | MATH-geometry                           |  -4.92567  |  -4.07634  |  -1.77914  |
> | MATH-counting_and_probability           |   2.45248  |  -0.666853 |   8.31177  |
> | MATH-number_theory                      |  -0.399989 |  -2.04254  |  -3.25859  |
> | GSM8K                                   |  -0.782802 |  -0.549927 |  -1.60607  |
> | BBH-geometric_shapes                    | -14.5554   |  -5.56531  |  -2.56755  |
> | BBH-salient_translation_error_detection |  -        |  -        |  -        |
> | BBH-word_sorting                        |   0.74345  |   0.850238 |  -5.06513  |
> | BBH-dyck_languages                      |       - |        - |                - |
>
> The table above shows the results of our experiment analyzing how the position of the most similar demonstrations (front, middle, or back of the prompt) impacts model performance across various tasks using Llama-3.1-8B. The performance is measured as the average percent change relative to the original order, with input lengths ranging from 1k to 64k.
>
> From the results, no clear universal trends emerged regarding the effect of demonstration positioning on performance across all tasks. For classification tasks, placing similar demonstrations at the front tended to hurt the model's performance more than placing them at the back. However, for non-classification tasks, no consistent pattern was observed.
>
> Importantly, these findings do not impact the overall conclusions of our experiment or paper, as our comparisons involve models evaluated with the same prompts across three random shuffles, ensuring robustness against ordering effects.

---

> > ### Author Response · Authors · 2024-12-02
> >
> > > **Classification tasks predominantly test models’ retrieval skills**
> >
> > In our paper, we argue that classification tasks predominantly evaluate a model’s retrieval capabilities. This is supported by our first experiment, which demonstrates that removing similar examples significantly degrades performance more than removing distant examples. This indicates that the model relies on similar examples for making predictions. If the model did not rely on these examples, the performance difference between removing similar and distant examples would be negligible.
> >
> > Additionally, our subsequent experiment shows that even when the label distribution remains consistent, replacing similar examples continues to negatively impact the performance of classification tasks. Importantly, we see no conflict between better illustration of the decision boundary and the model’s reliance on retrieval skills. A model that excels at retrieving similar demonstrations is inherently better at illustrating the decision boundary, as it leverages relevant examples to make more informed predictions.
> >
> > For very long prompts with numerous demonstrations, even when many similar examples are available to illustrate the decision boundary, models often struggle to maintain performance. This decline occurs because they are unable to effectively retrieve and utilize these similar demonstrations, highlighting a limitation in handling long-context inputs.
> >
> > We hope we have addressed your problem. Please let us know if you need more clarification.

---

> > > ### Comment · Reviewer_pTsV · 2024-12-03
> > >
> > > I thank authors for responding to my comments. However, I think your response reframed what you wrote in the paper. I would like to see deeper analysis and explanations. Based on your current response, suppose retrieval ability is a subset of global context understanding, how can this be explained in the context of the claims of the paper, such as "classification tasks benefit from more demonstrations" or "classification tasks predominantly test models’ retrieval skills?"
> > >
> > > The results on example ordering show a large impact on performance, with performance reduction of up to 14.5 points. Further analysis and results are needed to determine whether these major variations still support the main claims of the paper, particularly the two points mentioned above.

---

### Official Review · Reviewer_zS9W · 2024-11-04

**Soundness:** 3
**Presentation:** 3
**Contribution:** 3
**Rating:** 6
**Confidence:** 4

**Summary:**

The paper introduces Many-ICLBench, a new many-shot ICL benchmark. The analysis first dives into the trends of models across different lengths and categories of ICL tasks. Then, novel metrics, Retrieval Load Ratio and Global Context Index, are used to divide tasks into two categories. Finally, many existing open-source models are evaluated on the benchmark, revealing interesting insights.

**Strengths:**

- The paper builds upon previous works on many-shot in-context learning by extending the analysis to more types of tasks, such as summarization, translation, and reasoning tasks. These tasks provide a more holistic and realistic evaluation of long-context language models for ICL.
- The paper reveals interesting findings; in particular, the categorization of ICL tasks into retrieval vs. global context understanding can help practitioners in choosing which datasets to use during evaluation.
- Many-ICLBench may be a useful artifact to the community to test long-context language models.

**Weaknesses:**

- InfiniteBench also includes a diverse set of long-context language modeling tasks, such as QA, summarization, and ICL. There is no strong argument that Many-ICLBench is the first to “create a realistic long-context benchmark emphasizing retrieval and global context understanding skills.” What makes Many-ICLBench more appealing for users to test on over InfiniteBench?
- The findings can benefit from the inclusion of more SoTA long-context language models in the analysis, such as GPT-4/4o, Claude, and Gemini. For instance, one of the findings on the translation task is that the tested models do not benefit from the increasing number of demonstrations. However, Gemini was able to see improvements on a similar task. It would be useful to provide empirical evidence to show that the lack of improvement stems from the lack of multilingual capabilities of the model/model size.
- BM25 does not seem sufficient as a measure between two examples, as it only measures the lexical overlap between them. Using metrics such as BERTScore or the score from a dense retriever that can capture semantic similarity would make the “Retrieval Load Ratio” more convincing. Furthermore, math problems or other reasoning tasks seem unlikely to have lexical overlap even if they are using similar reasoning steps, whereas classification tasks are more likely to. The work could use more validation on this measure.
- In Appendix C, it’s shown that the quantized and unquantized versions exhibit similar trends, but the absolute number appears to differ significantly on certain tasks at long lengths: Llama 3.1 70B at 64k differs up to 6 points on Symbolic. The difference in the absolute value may affect the finding “the paradox of model size” in Sec 6 since the larger models are quantized and may have a lower absolute performance while the smaller models are not quantized.

Missing citations on the role of ICL:
* Sewon Min, Xinxi Lyu, Ari Holtzman, Mikel Artetxe, Mike Lewis, Hannaneh Hajishirzi, and Luke Zettlemoyer. 2022. Rethinking the Role of Demonstrations: What Makes In-Context Learning Work?. In Proceedings of the 2022 Conference on Empirical Methods in Natural Language Processing, pages 11048–11064, Abu Dhabi, United Arab Emirates. Association for Computational Linguistics.
* Jane Pan, Tianyu Gao, Howard Chen, and Danqi Chen. 2023. What In-Context Learning “Learns” In-Context: Disentangling Task Recognition and Task Learning. In Findings of the Association for Computational Linguistics: ACL 2023, pages 8298–8319, Toronto, Canada. Association for Computational Linguistics.

**Questions:**

- How is the average Pearson correlation coefficient calculated for Figure 1b?
- It seems from Figure 1a that Llama 3.1 70B Instruct is able to perform well on all tasks at 64k context length before dropping at the 128k input length. How would the Global Context Index change if the inputs were expanded to include 32k and 64k input lengths? Furthermore, do different models exhibit similar or different Global Context Index?

---

> ### Author Response · Authors · 2024-11-24
>
> Thank you for your valuable feedback. We aim to address your comments below:
>
> > **Comparision to InfiniteBench**
>
> We acknowledge the contributions of InfiniteBench and will include this work in the related work section of our camera-ready version. We do not claim to be the first to create such a benchmark. Compared to InfiniteBench, our work focuses on identifying suitable tasks and evaluating models' retrieval and global context understanding capabilities. Our benchmark provides two categories of tasks that allow users to conduct fine-grained evaluations of long-context models across varying context lengths—an aspect not covered by previous works. Furthermore, future research can leverage our methods to categorize tasks and datasets, enabling the quantification and characterization of advancements in long-context models.
>
> >**Gemini-1.5-Pro Results**
>
> We emphasize the importance of conducting experiments on open-weight models to support reproducibility. Closed-source models may become obsolete over time, limiting their utility for long-term research. We include the new Gemini-1.5-Pro results on our benchmark and the updated Table 2 in the following table:
>
> Gemini-1.5-Pro
> | Task                         | 1000        | 2000  | 4000  | 8000  | 16000  | 32000  | 64000  | 128000 | Avg   | Avg L  |
> |------------------------------|-------------|-------|-------|-------|--------|--------|--------|--------|-------|--------|
> | **Retrieval**               | 36.40       | 47.31 | 58.01 | 65.49 | 71.43  | 74.22  | 72.43  | 72.42  | 62.21 | 73.03  |
> | **Global Context Understanding** | 58.26       | 60.88 | 61.30 | 65.20 | 65.05  | 65.12  | 62.38  | 63.61  | 66.20 | 66.92  |
>
> Similar to other open-weight models on retrieval tasks, Gemini-1.5-Pro begins to show performance degradation beyond 32k. However, it is one of only three models (alongside Qwen-2-72B and GLM-Chat-9B) that demonstrate impressive retrieval capabilities beyond 64k and maintain performance at 128k. On global context understanding tasks, Gemini-1.5-Pro significantly outperforms other open-weight models, showcasing its ability to grasp the global context and effectively utilize all the demonstrations.

---

> > ### Author Response · Authors · 2024-11-24
> >
> > >**Embedding-based Approach**
> >
> > We use BM25 because it is one of the most widely used retrievers [1]. In Figure 12 from Appendix E, we employ SentenceTransformers (SBERT) for a new experiment where similar examples are replaced to compute the similarity scores. To ensure that performance loss is not caused by the absence of certain labels, we conduct an additional experiment in which the top k% most similar examples are replaced with the most distant examples that share the same labels.
> >
> > Although embedding-based approaches like SBERT can better capture semantics, the results from BM25 and SBERT exhibit the same trend: classification tasks consistently show a higher retrieval load ratio compared to non-classification tasks.
> >
> > The new experiment results are presented in Figure 12 and Figure 13 from Appendix E, along with the table below:
> >
> > **Retrieval Load Ratio Results for Original and Replacement Experiments with BM25 and SBERT**
> > | Task                                    |   Remove (BM25) |   Replace (BM25) |   Replace (SBERT) |
> > |:----------------------------------------|----------------:|----------------:|------------------------:|
> > | banking77                               |        1.37432  |        1.09297  |                1.1924   |
> > | dialogRE                                |        1.11089  |        1.07847  |                - |
> > | trec_50                                 |        1.32233  |        1.1724   |                1.16085  |
> > | clinc150                                |        1.53891  |        1.04783  |                1.08026  |
> > | MATH-algebra                            |        1.00061  |        1.01333  |                0.995086 |
> > | MATH-geometry                           |        1.01391  |        0.977575 |                1.03627  |
> > | MATH-counting_and_probability           |        1.02319  |        1.00746  |                0.952077 |
> > | MATH-number_theory                      |        0.981024 |        1.0278   |                0.965633 |
> > | GSM8K                                   |        0.997295 |        1.00486  |                0.998426 |
> > | BBH-geometric_shapes                    |        1.34485  |        1.34859  |                -  |
> > | BBH-salient_translation_error_detection |        1.01839  |        1.00197  |                - |
> > | BBH-word_sorting                        |        0.982197 |        0.996679 |                0.996264 |
> > | BBH-dyck_languages                      |        0.957303 |        1.02183  |                - |
> >
> > We excluded XLSUM because computing similarity among many news articles is computationally expensive. For SBERT, we excluded geometric_shapes, Dyck_languages, and dialogRE because the inputs of the first two tasks primarily consist of symbols and numbers, while the inputs for dialogRE are too lengthy for the retriever to be effective.
> >
> > The results from the original experiment still hold under both retrievers. Classification tasks exhibit a higher retrieval load ratio, while non-classification tasks have ratios closer to 1, suggesting that classification tasks rely more heavily on retrieving similar examples.
> >
> >
> > > **The paradox of model size**
> >
> > For classification tasks, both quantized and non-quantized Llama models perform similarly. As shown in Table 2 from Section 6, at 128k, the performance of Llama-3.1-70B degrades more significantly than that of Llama-3.1-8B. In the camera-ready version, we plan to include results from full-precision models. However, since most users lack the resources to run full-precision models, especially in the long-context setting, we believe our conclusions remain meaningful even when using 4-bit models.
> >
> > > **Correlation Calculation**
> >
> > For a single task, we collect performance data across all context lengths and calculate the correlation between them. To determine the correlation for a task category, we compute the average correlation coefficients of all tasks within that category.
> >
> > > **Global Context Index**
> >
> > We have included a new plot for Llama-3.1-70B between 2k and 64k in Appendix F, Figure 13. The global context index remains unchanged. Due to time constraints, we have not tested other models yet but plan to include their results in the camera-ready version.
> >
> > We hope we have addressed your concerns and kindly ask you to re-evaluate your assessment. Please feel free to reach out if you have any additional questions or require further clarification.

---

> > > ### Comment · Reviewer_zS9W · 2024-11-25
> > > **Response to Rebuttal**
> > >
> > > Thank you for addressing my comments, and I maintain my positive score.

---

### Official Review · Reviewer_QkrA · 2024-11-04

**Soundness:** 3
**Presentation:** 2
**Contribution:** 2
**Rating:** 5
**Confidence:** 4

**Summary:**

The paper aims to investigate the long context understanding capability of long context language models (LCLMs) via many-shot in-context learning (ICL). Specifically, the work proposes *(1)* a retrieval load ratio metric to identify tasks that requires retrieval of similar ICL examples to perform effectively, and *(2)* a global context index to identify tasks that need true global context understanding capability to perform well. Lastly, the author compiled ManyICLBench, a benchmark to assess LCLMs' retrieval skills and global context understanding skills separately.

**Strengths:**

- In general, the paper is well written. The discussion of related works is comprehensive and thorough.
- The paper is well motivated and targets the important gap of the lack of evaluation for LCLMs' true context understanding ability.
- This work presents an extensive set of experiments, offering great empirical insights for the community.

**Weaknesses:**

- It is unclear as to how Section ```4```. fits into the paper. How exactly does identifying the tasks that perform better/worse with more shots contribute to evaluating the global understanding capability of LCLMs?

- A number of prior works [1][2][3] have studied many-shot ICL in LCLMs. This work tries to provide a more comprehensive evaluation by adding tasks besides classification, however, the experiments do not include any closed-source API models.

- In the retrieval load experiments, removing the 10% similar ICL example would likely results in an absence of certain labels or an imbalanced ICL set with respect to the test input -- making the performance drop inevitable and might not be attributable to the reliance of retrieval skills. This also explains the results on non-classification tasks as they typically have a much larger output label space.

- To the best of my understanding, it seems that the retrieval skill discussed in the paper refers to the model's skill of inferring from similar input-output demonstrations to answer the test input -- which is not entirely the same as Needle-in-A-Haystack-style tasks that are explicitly about finding and retrieving phrases in the context. The retrieval skills in ICL might also involve a certain degree of understanding, instead of retrieval alone. Thus, the results might not be able to disentangle retrieval skills and global context understanding skills.


[1] Li et al., *Long-context llms struggle with long in-context learning*. 2024.

[2] Bertsch et al., *In-context learning with long-context models: An in-depth exploration*. 2024.

[3] Agarwal et al., *Many-shot in-context learning*. 2024.

**Questions:**

- Why is BM25 retriever adopted? Have you experiment with embedding-based approach that might captures the semantics better?
- Line ```259```: llama-3.1-7B --> 8B.
- This is a rather minor point -- but it might be better to move Figure ```6``` & ```7``` to the main context as they are referred in the main discussion.

---

> ### Author Response · Authors · 2024-11-24
>
> Thank you for your valuable feedback. We aim to address your comments below:
>
> >**How section 4 fits into the paper**
>
> Section 4 presents a preliminary study we conducted to explore how many-shot ICL enhances model performance across different task types. In this section, we aim to validate whether providing additional shots benefits models on various tasks. Intuitively, all tasks are expected to exhibit a non-decreasing trend as more shots are added. However, in Section 4, we observe that classification and non-classification tasks follow different trends when more shots are provided. We hypothesize that classification tasks with a positive trend primarily require models to retrieve similar examples, while non-classification tasks with a flat or negative trend demand that models understand each example in the prompt.
>
> This observation motivates us to systematically study why some tasks experience performance degradation with more shots while others do not. If the model struggles with non-classification tasks as the number of shots increases, is this due to its inability to retrieve similar demonstrations in the long prompt, or does it stem from a failure to leverage all the demonstrations to understand the task?
>
> >**Gemini-1.5-Pro Results**
>
> We emphasize the importance of conducting experiments on open-weight models to support reproducibility. Closed-source models may become obsolete over time, limiting their utility for long-term research. We include the new Gemini-1.5-Pro results and the updated Table 2 in the following table:
>
> Gemini-1.5-Pro
> | Task                         | 1000        | 2000  | 4000  | 8000  | 16000  | 32000  | 64000  | 128000 | Avg   | Avg L  |
> |------------------------------|-------------|-------|-------|-------|--------|--------|--------|--------|-------|--------|
> | **Retrieval**               | 36.40       | 47.31 | 58.01 | 65.49 | 71.43  | 74.22  | 72.43  | 72.42  | 62.21 | 73.03  |
> | **Global Context Understanding** | 58.26       | 60.88 | 61.30 | 65.20 | 65.05  | 65.12  | 62.38  | 63.61  | 66.20 | 66.92  |
>
> Similar to other open-weight models on retrieval tasks, Gemini-1.5-Pro begins to show performance degradation beyond 32k. However, it is one of only three models (alongside Qwen-2-72B and GLM-Chat-9B) that demonstrate impressive retrieval capabilities beyond 64k and maintain performance at 128k. On global context understanding tasks, Gemini-1.5-Pro significantly outperforms other open-weight models, showcasing its ability to grasp the global context and effectively utilize all the demonstrations.

---

> > ### Author Response · Authors · 2024-11-24
> >
> > > **New Retrieval Load Experiment**
> >
> > To ensure that the performance loss is not due to the absence of certain labels, we conducted an additional experiment where we replaced the top k% most similar examples with the most distant examples that share the same labels. This ensures that the label distribution of the new prompt remains identical to the original. The new retrieval load ratio is /frac{score_original}{ score_replace}. If the performance of a task degrades or its ratio increases, it indicates that the replacement of similar examples is the sole cause, highlighting the task's reliance on retrieval.
> >
> > We experimented with both BM25 and SentenceTransformer (SBERT) as retrievers, and the new experiment results are presented in Figure 12 and Figure 13 from Appendix E, along with the table below:
> >
> > **Retrieval Load Ratio Results for Original and Replacement Experiments with BM25 and SBERT**
> > | Task                                    |   Remove (BM25) |   Replace (BM25) |   Replace (SBERT) |
> > |:----------------------------------------|----------------:|----------------:|------------------------:|
> > | banking77                               |        1.37432  |        1.09297  |                1.1924   |
> > | dialogRE                                |        1.11089  |        1.07847  |                - |
> > | trec_50                                 |        1.32233  |        1.1724   |                1.16085  |
> > | clinc150                                |        1.53891  |        1.04783  |                1.08026  |
> > | MATH-algebra                            |        1.00061  |        1.01333  |                0.995086 |
> > | MATH-geometry                           |        1.01391  |        0.977575 |                1.03627  |
> > | MATH-counting_and_probability           |        1.02319  |        1.00746  |                0.952077 |
> > | MATH-number_theory                      |        0.981024 |        1.0278   |                0.965633 |
> > | GSM8K                                   |        0.997295 |        1.00486  |                0.998426 |
> > | BBH-geometric_shapes                    |        1.34485  |        1.34859  |                -  |
> > | BBH-salient_translation_error_detection |        1.01839  |        1.00197  |                - |
> > | BBH-word_sorting                        |        0.982197 |        0.996679 |                0.996264 |
> > | BBH-dyck_languages                      |        0.957303 |        1.02183  |                - |
> >
> > We excluded XLSUM because computing similarity among many news articles is computationally expensive. For SBERT, we excluded geometric_shapes, Dyck_languages, and dialogRE because the inputs of the first two tasks primarily consist of symbols and numbers, while the inputs for dialogRE are too lengthy for the retriever to be effective.
> >
> > The results from the original experiment still hold under both retrievers. Classification tasks exhibit a higher retrieval load ratio, while non-classification tasks have ratios closer to 1, suggesting that classification tasks rely more heavily on retrieving similar examples.
> >
> >
> > >**Comparision to Needle-in-A-Haystack**
> >
> > Needle-in-A-Haystack also requires a certain level of semantic understanding, as the model must comprehend the query to retrieve the key. Our retrieval tasks resemble a multi-key version of Needle-in-A-Haystack: the test example serves as the query, similar demonstrations act as the keys, and all non-relevant examples form the haystack. In Section 5, through retrieval load experiments, we demonstrate that the models primarily rely on their retrieval ability to solve these tasks.
> >
> > >**Embedding-based Approach**
> >
> > We use BM25 because it is one of the most widely used retrievers [1]. In Figure 12 from Appendix E, we employ SentenceTransformers (SBERT) for the new experiment, which replaces similar examples to compute the similarity scores. While embedding-based approaches like SBERT can better capture semantics, the results from BM25 and SBERT exhibit the same trend. All classification tasks consistently show a higher retrieval load ratio compared to non-classification tasks.
> >
> > > **Typos and Presentation Style**
> >
> > Thank you for your suggestions. We will revise our work accordingly.
> >
> > We hope we have addressed your concerns and kindly ask you to re-evaluate your assessment. Please feel free to reach out if you have any additional questions or require further clarification.
> >
> > [1] P. Zhao. et al., Retrieval-Augmented Generation for AI-Generated Content: A Survey, 2024

---

> > > ### Author Response · Authors · 2024-12-02
> > >
> > > As the discussion period deadline is approaching, please don’t hesitate to reach out if you need any additional clarification or further details on anything. We deeply appreciate your input and are happy to address any questions or concerns you may have!

---

> ### Comment · Reviewer_QkrA · 2024-12-02
> **Response to Author Rebuttals**
>
> Thank the authors for their response and additional experiments! My concerns are partially addressed. After reading other reviewers' comments (in particular, I share the concern of Reviewer ```pTsV``` on problem formulation) -- I believe the rating reflected by my original score is appropriate.

---

### Note · Authors · 2024-12-13

**Comment:**

We sincerely thank all the reviewers for their valuable insights and feedback. After careful consideration, we have decided to withdraw this work and continue refining it based on the suggestions provided.

**Withdrawal Confirmation:**

I have read and agree with the venue's withdrawal policy on behalf of myself and my co-authors.